# Optimal Training Positive Sample Size Determination for Deep Learning with a Validation on CBCT Image Caries Recognition

**DOI:** 10.3390/diagnostics14182080

**Published:** 2024-09-20

**Authors:** Yanlin Wang, Gang Li, Xinyue Zhang, Yue Wang, Zhenhao Zhang, Jupeng Li, Junqi Ma, Linghang Wang

**Affiliations:** 1National Center for Stomatology & National Clinical Research Center for Oral Diseases & National Engineering Research Center of Oral Biomaterials and Digital Medical Device & Beijing Key Laboratory of Digital Stomatology & NHC Key Laboratory of Digital Stomatology, Department of Oral and Maxillofacial Radiology, Peking University School and Hospital of Stomatology, Beijing 100080, China; 1610303129@pku.edu.cn (Y.W.); 1810303109@pku.edu.cn (X.Z.); 2School of Electronic and Information Engineering, Beijing Jiaotong University, Beijing 100044, China; 22120143@bjtu.edu.cn (Y.W.); 21120175@bjtu.edu.cn (Z.Z.); lijupeng@bjtu.edu.cn (J.L.); 3YOFO Medical Technology Co., Ltd., Hefei 230093, China; junqi.ma@yofomedical.com (J.M.); linghang.wang@yofomedical.com (L.W.)

**Keywords:** deep learning, training set, oral radiology, CBCT, dental caries

## Abstract

**Objectives**: During deep learning model training, it is essential to consider the balance among the effects of sample size, actual resources, and time constraints. Single-arm objective performance criteria (OPC) was proposed to determine the optimal positive sample size for training deep learning models in caries recognition. **Methods**: An expected sensitivity (P_T_) of 0.6 and a clinically acceptable sensitivity (P_0_) of 0.5 were applied to the single-arm OPC calculation formula, yielding an optimal training set comprising 263 carious teeth. U-Net, YOLOv5n, and CariesDetectNet were trained and validated using clinically self-collected cone-beam computed tomography (CBCT) images that included varying quantities of carious teeth. To assess performance, an additional dataset was utilized to evaluate the accuracy of caries detection by both the models and two dental radiologists. **Results**: When the number of carious teeth reached approximately 250, the models reached the optimal performance levels. U-Net demonstrated superior performance, achieving accuracy, sensitivity, specificity, F1-Score, and Dice similarity coefficients of 0.9929, 0.9307, 0.9989, 0.9590, and 0.9435, respectively. The three models exhibited greater accuracy in caries recognition compared to dental radiologists. **Conclusions**: This study demonstrated that the positive sample size of CBCT images containing caries was predictable and could be calculated using single-arm OPC.

## 1. Introduction

Caries is a chronic, progressive, bacterial-related oral disease widespread globally. According to the Global Status Report on Oral Health: Towards Universal Oral Health Coverage by 2030 [1] released by the World Health Organization (WHO) in 2022, nearly half of the global population will suffer from oral diseases, with dental caries being the most prevalent. The global prevalence of dental caries in deciduous and permanent teeth is 43% and 29%, respectively, affecting almost one-third of the population. As reported by the Fourth National Oral Health Epidemiological Survey in China [2], the rate of permanent tooth caries increases with age from 44.4% in the 15-year-old age group to 98.0% in the 65–74-year-old age group. If left untreated, dental caries may evolve into pulpitis, residual crowns, and roots or even result in tooth loss, negatively affecting aesthetics and chewing function. In severe cases, it may also contribute to other systemic problems [3,4]. Therefore, early diagnosis and treatment are crucial for preventing and controlling dental caries.

Diagnosing dental caries usually relies on clinical probing and X-ray examinations [5,6,7]. Two-dimensional bitewing and periapical radiographs, as well as three-dimensional cone-beam computed tomography (CBCT), are commonly used to assess the extent and depth of lesions and their relationship with the pulp. Two-dimensional radiographs are obtained with a low dose of radiation; however, they may result in anatomical image overlap, distortion, and deformation. Three-dimensional images can be used to determine the location and extent of a lesion by comparing the spatial relationship among three-dimensional anatomical structures [8,9]. However, three-dimensional images generate a higher level of radiation than two-dimensional images. Therefore, document no.172, titled “Guidelines for the Application of Evidence-Based Medicine for Oral and Maxillofacial Cone-Beam Computed Tomography”, published by the European Commission on Radiological Protection, explicitly states that CBCT should not be used exclusively for diagnosing caries. Caries should be carefully diagnosed if detected during CBCT examination for any other reason.

In recent years, with the development of deep learning neural networks, research on artificial intelligence (AI) in dental and medical imaging has progressed, leading to significant progress in diagnosing dental caries, periapical lesions, periodontitis, jaw cysts, etc. [10,11,12]. U-Net, YOLO, and ResNet are the most commonly used deep learning models. In dental images, these three models can segment structures such as teeth, pulp cavity, maxillary sinus, and mandibular canal. They can also be applied to segmental lesions, such as periapical and jaw cysts [13,14,15,16,17,18].

Studies have shown that deep learning neural networks perform well in diagnosing caries using 2D images, such as periapical radiographs [19,20,21], bitewing radiographs [22,23,24,25,26], and panoramic radiographs [27,28,29], with well-validated accuracy, sensitivity, and specificity. However, concerning 3D images, only one study [30] has been conducted on using CBCT images for caries diagnosis using ResNet and DenseNet. In this study, it should be noted that the training set determined for model training solely depended on the number of images collected, not on any other reasonable calculation. This implies that the number of images used in the training set could have been more optimal.

Data collection is a common step in the development of deep learning algorithms. Deep learning typically requires numerous annotated or labelled images for training. However, medical imaging data presents several limitations, including stringent patient confidentiality requirements, a scarcity of available data, a lack of standardized annotation protocols, significant costs, and the intensive human resources demanded for processing. At the same time, excessive data can also lead to the saturation of model performance, wasting memory and time resources. Ponnada et al. [31] found that in deep learning segmentation of brain MRI images from patients with multiple sclerosis, the performance of deep learning was strongly correlated with the number of samples. The results showed that when the number of training samples increased from 10 to 200, the performance of deep learning gradually improved; however, when the number of training samples increased from 200 to 800, there was no significant improvement. Therefore, during model training, it is essential to consider the balance between sample size and the available resources and time, and ideally to find a solution for determining the appropriate sample size for deep learning.

In this study, we proposed and evaluated a hypothesis for determining the optimal positive sample size for deep learning training sets using single-arm objective performance criteria. This was based on the application of U-Net, YOLOv5n, and CariesDetectNet models for caries recognition on CBCT images.

## 2. Materials and Methods

### 2.1. The Positive Sample Size in the Training Set

The single-arm objective performance criteria method, outlined in Formula (1), is frequently employed for calculating sample sizes in clinical settings.
(1)$$n=\frac{{\left[Z_{1-\frac{\alpha }{2}}\sqrt{P_{0}\left(1-P_{0}\right)}+Z_{1-\beta }\sqrt{P_{T}(1-P_{T})}\right]}^{2}}{({P_{0}-P_{T})}^{2}}$$

*n* represents the positive sample size; α represents the probability of type *I* error, and *β* represents the probability of type II error. *Z*_(1−*α*/2)_ represents the percentile of the standard normal distribution at the level of significance, which is usually set at 0.05 for both sides or 0.025 for one side; *Z*_(1−*β*)_ represents the percentile of the standard normal distribution for power, which is generally calculated at *β* = 0.1 (i.e., 90% power). *P*_0_ represents the clinically acceptable standard for the evaluation index, which can be formulated by reviewing professional literature or guidelines, evaluating current medical practices and technological levels, and discussing and communicating with experts. *P_T_* represents the expected value of the evaluation index, which is usually confirmed by referring to relevant historical studies or based on pre-experimental results; (*P_T_* − *P*_0_) represents the difference in overall parameters, that is, the difference between the expected value and the acceptable standard. Regarding the choice between two-sided and one-sided tests, a two-sided test was used if the experimental results were higher or lower than expected. A one-sided test was used if there was definite evidence that the experimental results would not be lower or higher than anticipated.

The number of positive samples is calculated by incorporating the estimated sensitivity, while the number of negative samples is calculated by incorporating the estimated specificity. In this study, we concerned with the number of positive samples; therefore, we first set sensitivity as the target value to estimate the optimal positive sample size for the training set. We determined P_0_ and P_T_ by reviewing the literature and evaluating current clinical caries diagnosis and treatment practices. Then, we determined the values of other parameters in the single-arm objective performance criteria, such as α and β. Next, we decided whether to use a two-sided or one-sided test. Finally, these parameters were substituted into the formula, and the single-arm objective performance criterion calculation formula was used to estimate the required positive sample size. Based on this estimation, we collected data. It is important to emphasize that this method serves only as an estimation tool, and the actual size of the training set may vary depending on the specific requirements of the task, such as data complexity and model structure.

Some studies have shown significant variation in the sensitivity of dentists’ diagnosis of dental caries, with sensitivity ranging from 0.35 to 0.56 for occlusal caries and 0.24 to 0.43 for interproximal caries [6,32]. Therefore, we set P_0_ to an intermediate value of 0.5 in this study. Based on the results of Ezhov et al. [30], the sensitivity of AI in diagnosing dental caries using CBCT was 0.6634; therefore, we set P_T_ to 0.6. We also set α and β at clinically commonly used values of 0.05 and 0.1, adopted a two-sided test, and applied these values to the single-arm objective performance criterion calculation formula. The final optimal positive training set size was determined, and 263 carious teeth were included in the training set.

### 2.2. Data Collection and Division

The hospital’s ethics committee approved this study, and the exemption of informed consent had been granted for retrospective study. All CBCT images were retrospectively collected from the Picture Archiving and Communication System (PACS) of the Department of Oral and Maxillofacial Radiology, Peking University School and Hospital of Stomatology, between January 2022 and October 2023.

The images were acquired with a NewTom VG/VGi CBCT unit (Quantitative Radiology, Verona, Italy) at 110 kVp and 1–5 mA and a scanning field of view of 160 × 160/120 × 80 mm^2^. While being examined, the patient took a standing position and were asked to keep the intercuspal position to prevent motion artifact. The patients’ Frankfurt horizontal planes were parallel to the ground. The inclusion criteria for CBCT images were as follows: (1) patients aged between 20 and 50 years with fully developed teeth and jaws, (2) both the maxillary and mandibular dentition fully captured in one CBCT image, (3) the presence of at least one carious tooth in the CBCT image, and (4) carious teeth without any dental treatment or crown restoration. The exclusion criteria were as follows: (1) edentulous patients, (2) severely distorted or blurred images, and (3) the presence of structures or diseases in the CBCT images that may impact the diagnosis.

Translucent areas in the teeth were classified as carious lesion when the following imaging features were met: (1) disruption of enamel margin continuity, extensive of low-density shadows and areas of decreased density that propagate along the enamel-dentin junction; (2) reduced density of enamel margins and low-density shadows confined to the enamel. All images were initially diagnosed as carious teeth by a dentist with over two years of clinical expertise, and these diagnoses were subsequently validated and reviewed by an experienced oral and maxillofacial radiologist with a minimum of 20 years of professional practice.

Dataset 1 was our initial dataset, comprising 259 CBCT images of 376 carious teeth, and was used to train the U-Net and YOLOv5n models. Dataset 2 expanded on Dataset 1, containing 425 CBCT images of 626 carious teeth, and was used to train the CariesDetectNet model. Finally, 37 carious teeth from 25 patients were selected as the final test group (FTG) to compare the diagnostic performances of the three best-performing models with those of dental radiologists. Thus, the total dataset included 450 CBCT images from 450 patients including 663 caries.

In the total dataset, 246 cases (54.7%) were male and 204 cases (45.3%) were female. Among these, 155 cases (34.4%) belonged to the 18–29 age group, 151 cases (33.6%) belonged to the 30–39 age group, 61 cases (13.6%) belonged to the 40–49 age group, and 83 cases (18.4%) belonged to the age group greater than 50 years old. Regarding the number of carious teeth, 267 patients (59.3%) had one carious tooth, 117 patients (26.0%) had two caries, 46 patients (10.2%) had three caries, and 20 patients (4.5%) had more than three caries. As for the location of the caries, 97 caries (14.7%) were in the anterior teeth, 188 caries (28.3%) were in the premolar teeth, and 378 caries (57.0%) were in the molar teeth.

Table 1 shows the specific numbers of carious teeth and patients in the three models’ training, validation, and testing sets. To avoid biased evaluation of the model performance due to data leakage, we strictly followed the norms of data division to ensure that all the data (including CBCT images, teeth, caries, etc.) of the same patient were used only for training, validation, or testing individually; that is, if a patient had three carious teeth, all three carious teeth of this patient would be assigned to the same group and not used in any other groups.

#### 2.2.1. Data Division of U-Net

Dataset 1 was divided into three mutually exclusive sets: training, validation, and testing. Following a ratio of 8:1:1 for the training, validation, and testing sets, we chose 300 carious teeth from 200 patients as the total training set, each of which may contain multiple caries; we chose 38 carious teeth from 29 patients as the total validation set and another 38 carious teeth from 30 patients as the total testing set.

The 300 carious teeth in the training set were divided into 12 groups of 25 carious teeth each. One group was randomly selected from 12 groups to form an initial training set. The training set was subsequently expanded by randomly incorporating data from the remaining groups. Consequently, the initial training set comprised CBCT images with 25 carious teeth, with subsequent sets increasing with CBCT images containing 25 carious teeth as a gradient until a total dataset containing 300 carious teeth was reached. Using the same method, both the initial validation and testing set began with three carious teeth from one or multiple CBCT images and were then expanded with an increment of three carious teeth each time for the corresponding validation and testing sets, finally reaching twelve carious teeth in the total validation and testing sets, respectively. Thus, each training set had a corresponding validation and testing set.

#### 2.2.2. Data Division of YOLOv5n

The number of carious teeth in the total training, validation, and testing sets for the model establishment with YOLOv5n was the same as that for U-Net. The dataset grouping methodology was also similar to that for U-Net except for the increment of each training set to 50 carious teeth and the validation and testing sets with an increment of 6 carious teeth each time in consideration of YOLO’s suitability for large-scale datasets. This resulted in six groups, each comprising a training set and corresponding validation and testing sets (Table 1).

#### 2.2.3. Data Division of CariesDetectNet

Dataset 2 was divided into three sets: a comprehensive training set, a validation set, and a testing set, following an 8:1:1 distribution. The training set included 500 carious teeth from 323 patients, and the validation and testing sets included 63 carious teeth from 50 and 52 patients, respectively. The gradient of the training set was established for the 50 carious teeth. The dataset grouping methodology was aligned with that of U-Net. Consequently, we obtained ten groups, each comprising a training set and its corresponding validation and testing sets.

### 2.3. Data Normalization and Annotation

The CBCT images were normalized to achieve an isotropic resolution of 0.3 × 0.3 × 0.3 mm^3^. Additionally, the intensity values of each CBCT scan were constrained within the range of [−999, 3000].

#### 2.3.1. Data Annotation for U-Net

A dental radiologist with more than three years of clinical experience conducted semi-automatic labelling of carious teeth in Dataset 1 and FTG, utilizing ITK-SNAP software (V. 3.8, Cognitica, Philadelphia, PA, USA) for slice-by-slice fine adjustments (refer to Figure 1 and Figure 2). All CBCT images and their corresponding labelled data were initially stored in NIFIT format and converted into PNG files, each with a 512 × 512 pixels resolution and an 8-bit depth. Subsequently, all slices featuring carious lesions were extracted from the CBCT images and their corresponding labelled data. Each CBCT image and corresponding labelled data were assigned independently to the same identifier. Finally, slices from each patient underwent random augmentation, employing two of three techniques: affine transform stretching, random rotation, and mirroring.

#### 2.3.2. Data Annotation for YOLOv5n

The CBCT images in Dataset 1 and FTG were transformed into the NIFIT format utilising ITK-SNAP software V. 3.8 for preservation. Subsequently, the CBCT axial images were converted into PNG format files, each with a resolution of 512 × 512 pixels and an 8-bit depth, utilising DICOM Converter software (V.1.12.0 DICOM Apps). All slices containing caries were selected from the CBCT axial images in Dataset 1 and the FTG, and each selected CBCT axial image was assigned an independent number, resulting in 2357 images. The same dental radiologist employed the LabelImg software (V.1.8.6 Tzutalin, 2015. LabelImg. Git code.) to annotate all CBCT axial slices featuring caries-affected regions (Figure 3), with the category labelled as “Caries”.

#### 2.3.3. Data Annotation for CariesDetectNet

The DICOM-formatted CBCT images of carious teeth in Dataset 2 and FTG were loaded into MilPai software (V.2.4 YOFO Medical, Hefei, China), which is professionally designed software specifically suited for dental applications. The software subsequently produced the corresponding panoramic images based on the CBCT images. The same dental radiologist employed circular annotations to mark all the carious teeth on the panoramic images, situating the centre of each circle at the epicentre of the carious area. Adjustments were made such that the pinpoints were accurate for carious lesions in the three-dimensional CBCT images (Figure 4).

### 2.4. Architecture of the Three Models

#### 2.4.1. Architecture of U-Net

The basic structure of U-Net is shown in Figure 5 and consists of an encoder and a decoder. The encoder comprises three modules, each containing two convolutional layers and a pooling layer. The decoder is composed of three modules. To improve the performance, a deep supervision method was adopted to train the network, with an auxiliary classifier added to the hidden layer of the decoder as a branch for supervised learning. U-Net was trained using the early stopping method with the following parameter settings: epoch = 360, batch size = 8, and learning rate (lr) = 0.0001. Dice Loss and Cross Entropy Loss (CE) were the loss functions. Twelve training sets and their corresponding validation sets were input into U-Net for training, and the best-performing network parameters were saved to evaluate their performance on the corresponding testing sets.

#### 2.4.2. Architecture of YOLOv5n

Figure 6 shows the basic structure of YOLOv5n, which includes an input end, Backbone, Neck module, and output end. The Backbone used for feature extraction comprised the focus, Cross Stage Partial (CSP), and Spatial Pyramid Pooling Fast (SPPF) modules. For the Neck module, we adopted a method that combined a Feature Pyramid Networks (FPN) and a Path Aggregation Network (PAN) to fuse features. Genetic algorithms were used to optimize the hyperparameters of YOLOv5n, and CIoU_LOSS was employed as the loss function for the bounding boxes.

#### 2.4.3. Architecture of CariesDetectNet

CariesDetectNet consists of a tooth detection model and a three-dimensional caries detection model. The leading architecture of the panoramic-image tooth-detection model is based on YOLOv4 (Figure 7), and its basic structure is similar to that of YOLOv5. The three-dimensional caries detection model borrowed the idea of ResNet, adopting a convolutional structure that serially connected 3D Conv and ResBlock modules. This is an improved version of the 3D-Conv-ResNet network, and its overall structure is illustrated in Figure 8. By adding residuals after the convolutional blocks to obtain more effective feature extraction, the number of channels is decreased and the depth is increased as the data passes through the network. The network was connected to two linear layers for classification. The final output of the model consisted of five parameters: the centre point coordinates (x, y, z) of the caries, radius r, and probability of caries.

For details of the code of the deep learning network, please refer to Appendix A.

### 2.5. Test with the FTG

Slices of the CBCT images from the FTG were input into the best-performing model obtained from the training process, and the output was the carious teeth detected by the model.

Two dental radiologists with at least three years of experience in CBCT imaging diagnosis were used as the comparator group to estimate dental caries in the CBCT images. They were blinded to the patient’s history. Each participant independently marked the carious teeth in 24 CBCT images; 37 teeth showed caries of different degrees of severity, and 665 were negative. The red box indicates carious teeth, the blue box indicates possible carious teeth, and no markings indicate non-carious teeth. The inter-observer variance was assessed using Cohen’s kappa test for the two dental radiologists. According to Landis and Koch (1977), agreement was interpreted as being slight (0.00–0.20), fair (0.21–0.40), moderate (0.41–0.60), substantial (0.61–0.80), almost perfect (0.81–0.99), or ideal (1.00).

### 2.6. Performance Metrics

Diagnostic accuracy, sensitivity, specificity, positive predictive value (PPV), negative predictive value (NPV), precision, recall, receiver operating characteristic (ROC) curve, and area under the Receiver Operating Characteristic (ROC) curve (AUC) of the test dataset were assessed. The chi-square test was utilized to analyse the differences in caries recognition between dental radiologists and the three models. Chi-squared tests and Cohen’s kappa statistics were conducted using SPSS software v23.0 (SPSS Inc., Chicago, IL, USA). Statistical significance was set at *p* < 0.05.

## 3. Results

Each model was trained independently. Table 2, Table 3 and Table 4 present the performance metrics of the three models in their corresponding testing sets following training with varying numbers of caries in the training sets. It was observed that the rate of performance improvement plateaued when the number of carious teeth in the training sets for the three models reached 200, 250, and 300, respectively. Specifically, when the number of carious teeth in the training sets reached 250, 300, and 300, the corresponding accuracies were 0.9299, 0.9868, and 0.9731. Figure 9 shows the ROC curves for YOLOv5n and CariesDetectNet.

Table 5 presents the performance results of the three models using the FTG, indicating their effectiveness after training with varying numbers of carious teeth in the training sets. U-Net and YOLOv5n exhibited optimal performance when trained with 250 carious teeth (Groups 10 and 5, respectively), indicating that these models are well suited for dental caries identification under the given conditions. CariesDetectNet exhibited minimal fluctuations in accuracy and specificity, with differences not exceeding 0.1. However, notable variability in the sensitivity of CariesDetectNet was observed across the groups, ranging from 0.6603 to 0.8490, primarily influenced by the number of carious teeth in the training set, with the peak sensitivity achieved at 250 carious teeth. By consolidating the results from Table 4 and Table 5, it is evident that CariesDetectNet performed optimally with training sets consisting of either 250 or 300 carious teeth. The number of dental caries included in the training set to achieve peak performance in these models mirrored the calculated 263 carious teeth. This suggests that the proposed method for estimating the positive sample size based on the single-arm objective performance criteria offers valuable insights for deep learning in identifying dental caries in CBCT images.

Table 5 presents the accuracy, sensitivity, specificity, PPV, NPV, and Youden Index of the three models and the dental radiologists in the FTG. The three models exhibited minimal differences in performance, with U-Net marginally outperforming the other models. Notably, U-Net achieved the highest Youden Index, indicating superior overall diagnostic performance. While the three models exhibited marginally higher accuracy and specificity compared to the dental radiologists, they also displayed lower sensitivity to varying degrees. This finding suggests a heightened risk of misdiagnosis associated with the models when compared to the assessments made by dental radiologists.

The Cohen’s kappa value was 0.72, indicating substantial agreement between the two dental radiologists.

## 4. Discussion

A positive sample size within the training set was pivotal in shaping the model’s performance. Theoretically, a small training set can lead to model overfitting and a lack of generalization ability [33], whereas a more extensive set typically results in improved deep-learning model performance. In medical imaging, although there are some publicly available medical image datasets, the usability of medical images may still be somewhat limited owing to privacy, legal, and ethical constraints, as well as the specific needs of the data for different studies and applications. The time-consuming nature of professional annotation increases the complexity. Furthermore, an excessive volume of data may not necessarily enhance the model performance [34]; instead, it can lead to unnecessary strain on computational resources and time constraints. When determining an appropriate positive sample size for training, it is essential to carefully consider the model’s requirements alongside practical considerations related to resource availability and time constraints. Therefore, determining the optimal training sample size for deep learning neural networks is crucial in achieving effective model performance. The single-arm objective performance criterion is a method for calculating sample size used in clinical diagnostic trials. Its fundamental principle involves determining the necessary positive sample size based on a predetermined outcome rate. To establish a deep-learning-based CBCT image caries detection model and evaluate its diagnostic performance, a diagnostic test with statistical testing was used as the goal, which met the calculation objective of the single-arm objective performance criteria. In deep learning, the training process involves both training and self-testing. In each iteration, the model utilizes a portion of the data for training while simultaneously employing another portion for self-testing and adjustment. The model gradually optimizes its performance through continuous parameter adjustments and achieved the best performance. From this perspective, the training set can also be considered a “testing set” for deep learning models, which fulfils the application conditions of the single-arm objective performance criteria. Based on this understanding, we propose applying single-arm objective performance criteria to estimate a positive sample size in deep learning.

Currently, positive sample size determination for artificial intelligence mainly involves estimation based on model algorithm characteristics (a priori method) and curve fitting methods based on empirical data to model and extrapolate the algorithm performance (a posteriori method). Curve-fitting methods include learning and linear curve fitting. However, both approaches have limitations. A sample size estimation method based on model algorithm characteristics may struggle to capture the complexity of specific deep learning tasks, particularly in high-dimensional environments with substantial intragroup variability [35]. It must be adjusted according to the bias-variance trade-off of different algorithms [36] and the function generation mechanism [37]. The linear curve fitting method relies heavily on the algorithm, the number of features, and their distribution when evaluating effectiveness [38,39,40,41]. However, when estimating the minimum sample size, the linear curve-fitting method needs to be revised, as adding any amount of data leads to a nonzero increase in the training dataset [40]. Moreover, these methods are designed for specific tasks, datasets, and models and have yet to be validated for their effectiveness in diverse tasks and image types within a healthcare setting.

Although existing methods for determining positive sample sizes in deep learning training sets have not been widely implemented in dentistry, more research is needed on positive sample size calculation methods specifically designed for deep learning training sets using dental images. A few researchers have recognized the significance of positive sample size in deep learning and have initiated studies exploring this area.

Semih et al. [42] combined image processing and deep learning algorithms to detect tooth numbering issues in panoramic radiographs and studied the impact of data volume on model performance. They collected 3000 panoramic radiographs and used four different datasets, including 1000, 1500, 2000, and 2500 panoramic radiographs, to train the YOLOv4 algorithm to investigate the relationship between the amount of data used in image processing algorithms and model performance and tested them on a testing set containing 500 panoramic radiographs. The metrics of the F1-score, sensitivity, precision, and recall rate obtained from the models were compared. As the amount of data in the training set increased, the model’s performance also improved, and the model trained with 2500 data points showed the highest performance among all the trained models. The results of this study indicate that the size of the training dataset is crucial for tooth counting and that larger samples are more reliable.

Ying et al. [43] developed deep learning models based on the U-Net and Trans-U-Net for caries segmentation. Using the single-arm objective performance criteria, the study estimated the positive sample size in the testing set with assumed minimum standards of expected sensitivity (p) and clinically acceptable sensitivity (p0) of 0.80 and 0.60, respectively—the resulting sample size for the testing set comprised of 55 teeth with caries. In total, 154 periapical radiographs were collected for the experiment, with 40 designated as the testing set. Different numbers of the original caries images (20, 40, 80, 100, and 113) were augmented to a uniform 800 images to train the network. As the number of original images increased, network performance gradually improved and plateaued at approximately 80 carious teeth. The network trained with 80 carious teeth showed a sensitivity of 0.94, specificity of 0.92, average Dice similarity coefficient of 0.7487, and average pixel classification accuracy of 0.7443 for caries detection. The study findings indicate that while the performance of deep learning models is enhanced with increased caries samples, an optimal positive sample size exists for the training set, beyond which the model’s performance does not significantly improve.

Similar to Semih and Ying’s study, the present study addressed the issue of positive sample size in deep-learning training sets. This study introduced the application of single-arm objective performance criteria to determine the positive sample size for a deep learning training set, exemplifying this by focusing on the detection of dental caries in CBCT images and validating it using three models: U-Net, YOLOv5n, and CariesDetectNet. These three models achieved optimal network performance when the number of teeth with caries in the training set was 250, 250, and 250/300, which aligned closely with the calculated optimal positive sample size. This highlights the value of the proposed method for estimating positive sample sizes using single-arm objective performance criteria in the context of deep learning for identifying dental caries in CBCT images.

U-Net is a widely utilized and highly effective convolutional neural network architecture, particularly suitable for image segmentation tasks. In the context of CBCT image caries recognition, U-Net’s ability to accurately segment the caries-affected regions within the complex anatomical structures of the jaw is crucial. YOLOv5n is a lightweight variant within the YOLO (You Only Look Once) series of object detection models, typically chosen for small object detection due to its balanced speed and accuracy. Its rapid and efficient nature makes YOLOv5n an attractive choice for caries detection in CBCT images. Its single-stage detection method allows it to directly predict bounding boxes and class probabilities from input images, significantly reducing computational complexity. Additionally, YOLOv5n is capable of detecting multiple instances of caries within a single image. CariesDetectNet consists of a panoramic dental image detection model and a 3D caries detection model. Although U-Net and YOLOv5n can employ 2D image processing models to analyse 3D objects, they necessitate that each slice of the biomedical imaging be individually input into the model for training, which may result in inefficiencies. To enhance efficiency, CariesDetectNet utilizes 3D Convolutions (3D Conv) for classification and feature extraction of CBCT images, enabling a more efficient recognition of caries. In conclusion, the selection of U-Net, YOLOv5n, and CariesDetectNet was predicated on their proven performance, suitability for the task of caries recognition on CBCT image, and their complementary properties that improve the overall accuracy, precision, and efficiency of our proposed method.

Among the three models, U-Net outperformed YOLOv5n and CariesDetectNet in terms of accuracy, sensitivity, specificity, PPV, and NPV by 0.9899, 0.9130, 0.9997, 0.9968, and 0.9918, respectively. Additionally, it required the smallest positive sample size, indicating superior caries detection performance using CBCT images. The sensitivity and PPV were significantly higher than those of the other models, suggesting that U-Net had a lower rate of missed diagnoses than other models.

Most studies have used deep learning to diagnose dental caries using two-dimensional images such as periapical and bitewing radiographs. Lee et al. [20], Chen et al. [21], and Cantu et al. [24] successfully developed deep-learning neural networks to detect caries on periapical and bitewing radiographs automatically. Bayraktar et al. [25] and Mohamed et al. [44] have utilized deep-learning neural networks to screen and detect caries in panoramic radiographs. For CBCT images, Matvey et al. [30] proposed an AI system based on a convolutional neural network (CNN) that includes a region of interest (ROI) localization module (for tooth and jaw segmentation), a tooth localization and counting module, a periodontitis module, and a periapical lesion localization module, and trained these modules with 1346 CBCT images. This AI system had a sensitivity of 0.7285 and a specificity of 0.9953 with the caries localization module. Compared with this caries localization module, the sensitivity and specificity in the present study using U-Net reached 0.9130 and 0.9997, respectively, which is much better than the method proposed by Matvey et al. [30].

In clinical practice, some patients were scanned with large-view CBCT for orthodontic and orthopaedic reasons, and on these CBCT images, dentists focus more on the overall relationship between the jaws and the teeth than on the possible lesions in the individual teeth. The purpose of this study was to provide dentists with a reference for the initial screening of caries in large-field-view CBCT images so that patients can be diagnosed and treated in time.

The sensitivity of caries diagnosis is closely related to the size and depth of the carious lesion. Enamel caries are more difficult to diagnose than dentinal caries and reduce the overall diagnostic sensitivity of caries. Since, in the present study, we did not know beforehand the status of carious lesions in the patients, the P0 of 0.5 and PT of 0.6 were set based on the generally accepted level of previous research and clinical practice.

There are some limitations to this study. (1) The data source is limited to only one brand of machine, which may not be universally applicable. (2) The study focused solely on validating the effectiveness of the optimal positive sample size estimation method using three common deep learning models. Further validation is necessary to establish its universality. (3) The study exclusively utilized CBCT images for dental caries diagnosis without incorporating other clinical information. (4) There were insufficient cases of certain specific types or specific conditions, and this potential diversity limitation may have an impact on the results of the study. (5) The three models have not yet been implemented in clinical practice; hence, their real-world impact on dental clinics should be assessed thoroughly.

## 5. Conclusions

This study introduced a novel approach based on single-arm objective performance criteria in diagnostic testing, aimed at determining the optimal positive sample size for deep learning to diagnose dental caries in CBCT images. The viability of this calculation method was verified using the U-Net, YOLOv5n, and CariesDetectNet models. Furthermore, this research evaluated the efficacy of deep learning models in identifying and diagnosing dental caries on CBCT images, providing valuable insights into calculating the optimal positive sample size for deep learning applications and their clinical implications.

## Figures and Tables

**Figure 1 diagnostics-14-02080-f001:**
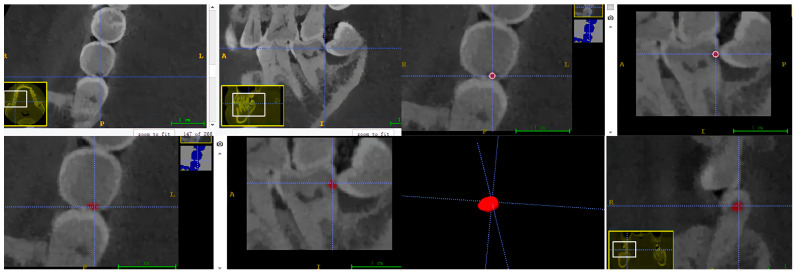
Semi-automated annotation process using ITK-SNAP software V. 3.8. The red indicates carious lesion.

**Figure 2 diagnostics-14-02080-f002:**
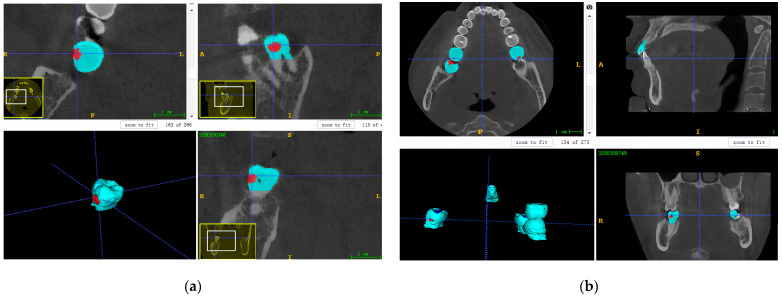
The manual annotation results. The red indicates carious lesions and the blue indicates the carious teeth. (**a**) The annotation result of a single tooth; (**b**) the annotation result of multiple teeth.

**Figure 3 diagnostics-14-02080-f003:**
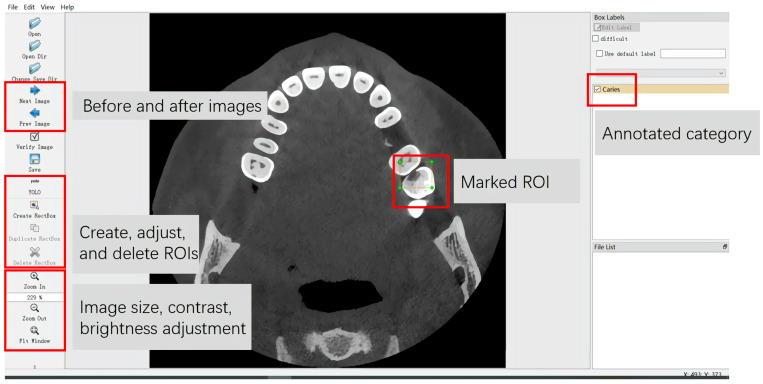
Annotation interface of YOLOv5n. The red boxs and gray squares indicate the meaning of keys.

**Figure 4 diagnostics-14-02080-f004:**
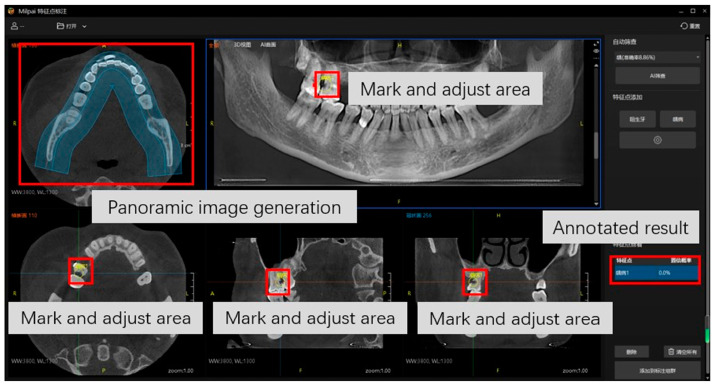
Annotation interface of MilPai. The red boxs and gray squares indicate the meaning of keys.

**Figure 5 diagnostics-14-02080-f005:**
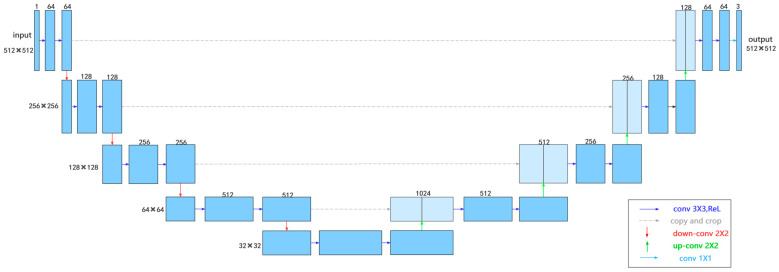
Structure of the U-Net.

**Figure 6 diagnostics-14-02080-f006:**
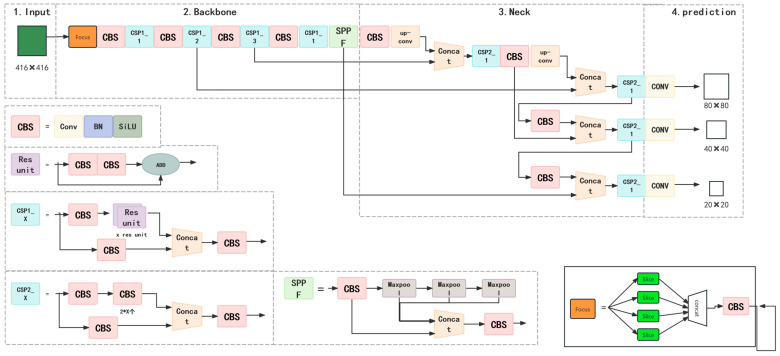
Structure of theYOLOv5n. The pink rectangles represent the major operations or layers within the YOLOv5n model. The purple rectangles is res unit, which solves the problem of gradient explosion. The blue rectangles represent Cross Stage Partial, which is a backbone network used to enhance the learning ability of CNNs. The green rectangles represent SPPF, which is used to address the limitation of fixed-size inputs in convolutional neural networks (CNN) and allows the network to process input images of arbitrary size. The orange rectangles rectangle the contact module that can help the deep learning network integrate information from different layers or different sources, thereby improving the network’s expressive power and performance.

**Figure 7 diagnostics-14-02080-f007:**
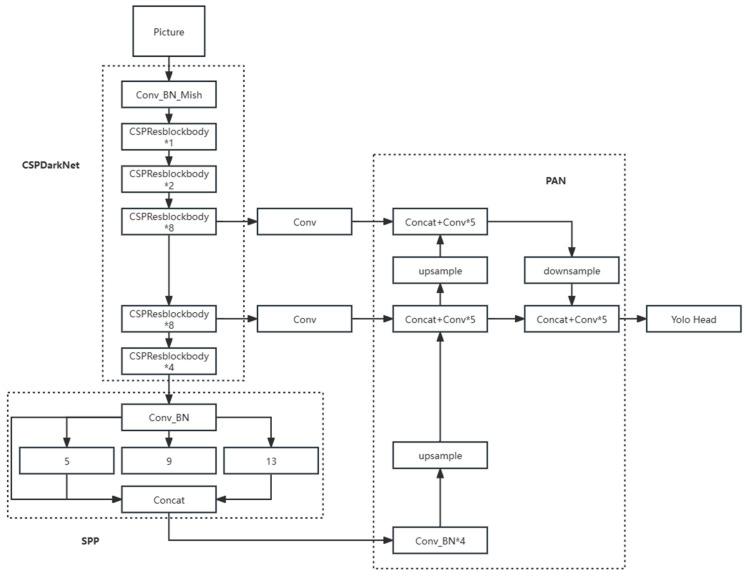
Structure of theYOLOv4. ‘n’ represents the number, and ‘*’ represents multiplication sign. ‘*n’ represents the module containing n identical structures.

**Figure 8 diagnostics-14-02080-f008:**
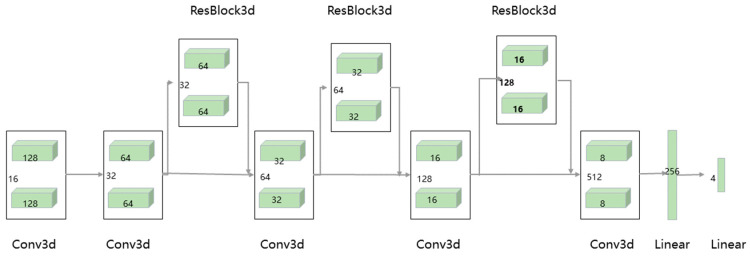
Structure of the 3D-Conv-ResNet network.

**Figure 9 diagnostics-14-02080-f009:**
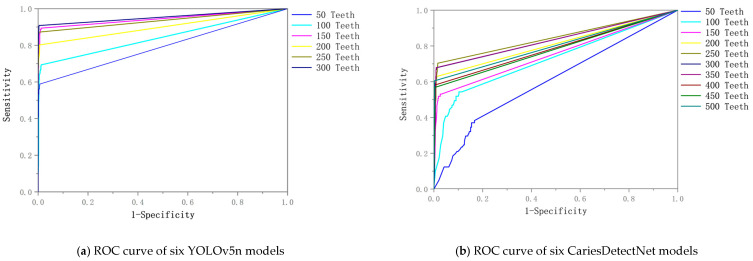
ROC curve of six YOLOv5n (**a**) and CariesDetectNet (**b**) models.

**Table 1 diagnostics-14-02080-t001:** Data division of U-Net, YOLOv5n, and CariesDetectNet.

		Number of Training Sets			Number of Validation Sets			Number of Testing Sets		
		Patients	Carious Teeth	Healthy Teeth	Patients	Carious Teeth	Healthy Teeth	Patients	Carious Teeth	Healthy Teeth
U-Net	Group 1	14	25	364	2	3	49	2	3	53
	Group 2	25	50	625	5	6	117	4	6	93
	Group 3	39	75	1011	7	9	144	6	9	138
	Group 4	49	100	1278	9	13	193	9	13	217
	Group 5	64	125	1692	10	16	249	11	16	265
	Group 6	85	150	2127	12	19	301	14	19	352
	Group 7	97	175	2503	15	22	378	16	22	407
	Group 8	117	200	3053	18	25	463	19	25	498
	Group 9	135	225	3517	21	28	534	21	28	557
	Group 10	156	250	4045	23	31	583	24	31	640
	Group 11	180	275	4664	26	34	666	26	34	696
	Group 12	200	300	5167	29	38	751	30	38	812
YOLOv5n	Group 1	25	50	625	5	6	117	4	6	93
	Group 2	49	100	1278	9	13	193	9	13	217
	Group 3	85	150	2127	12	19	301	14	19	352
	Group 4	117	200	3053	18	25	463	19	25	498
	Group 5	156	250	4045	23	31	583	24	31	640
	Group 6	200	300	5167	29	38	751	30	38	812
CariesDetectNet	Group 1	25	50	625	5	6	117	4	6	93
	Group 2	49	100	1278	9	13	193	9	13	217
	Group 3	85	150	2127	12	19	301	14	19	352
	Group 4	117	200	3053	18	25	463	19	25	498
	Group 5	156	250	4045	23	31	583	24	31	640
	Group 6	200	300	5167	29	38	751	30	38	812
	Group 7	236	350	6066	33	44	863	35	44	933
	Group 8	268	400	6885	39	50	1011	40	50	1043
	Group 9	291	450	7483	44	56	1121	46	56	1165
	Group 10	323	500	8344	50	63	1233	52	63	1289

**Table 2 diagnostics-14-02080-t002:** The accuracy, sensitivity, specificity, PPV, NPV, F1-score, DSC, IoU, Precision, and Recall values of the 12 U-Net models.

	Acc	Se	Sp	PPV	NPV	F1-Score	DSC	IoU	Precision	Recall
Group 1	0.9049	0.3548	0.9784	0.6875	0.9190	0.4680	0.6208	0.5474	0.6875	0.3548
Group 2	0.9486	0.5416	0.9912	0.8666	0.9537	0.6666	0.6817	0.5976	0.8667	0.5416
Group 3	0.9412	0.4090	0.9962	0.9183	0.9422	0.5660	0.6876	0.6136	0.9183	0.4090
Group 4	0.9598	0.5309	0.9962	0.9230	0.9615	0.6741	0.7251	0.6048	0.9230	0.5309
Group 5	0.9537	0.6285	0.9827	0.7652	0.9673	0.6901	0.7398	0.696	0.7652	0.6285
Group 6	0.9720	0.7459	0.9922	0.8961	0.9776	0.8141	0.8262	0.747	0.8961	0.7459
Group 7	0.9729	0.7407	0.9958	0.9467	0.9749	0.8311	0.8364	0.7621	0.9467	0.7407
Group 8	0.9817	0.8167	0.9980	0.9761	0.9821	0.8893	0.9165	0.8618	0.9761	0.8167
Group 9	0.9849	0.8446	0.9982	0.9780	0.9855	0.9065	0.9052	0.8449	0.9780	0.8446
Group 10	0.9929	0.9307	0.9989	0.9889	0.9932	0.9590	0.9435	0.9008	0.9889	0.9307
Group 11	0.9875	0.8790	0.9986	0.9853	0.9877	0.9291	0.9146	0.8636	0.9853	0.8790
Group 12	0.9899	0.9060	0.9987	0.9867	0.9902	0.9447	0.9116	0.8566	0.9867	0.9060

**Table 3 diagnostics-14-02080-t003:** The accuracy, sensitivity, specificity, PPV, NPV, F1-score, AUC, Precision, and Recall of the YOLOv5n models and CariesDetectNet models.

		Acc	Se	Sp	PPV	NPV	F1-Score	AUC	Precision	Recall
YOLOv5n	Group 1	0.9541	0.5753	0.9929	0.8936	0.9580	0.7000	0.7926	0.8936	0.5753
	Group 2	0.9618	0.6935	0.9866	0.8269	0.9721	0.7543	0.8429	0.8269	0.6935
	Group 3	0.9794	0.8940	0.9877	0.8766	0.9896	0.8852	0.9452	0.8766	0.8940
	Group 4	0.9764	0.7596	0.9938	0.9076	0.9809	0.8271	0.9003	0.9076	0.7596
	Group 5	0.9841	0.8540	0.9970	0.9669	0.9856	0.9069	0.9456	0.9669	0.8540
	Group 6	0.9868	0.8964	0.9958	0.9551	0.9897	0.9248	0.9535	0.9551	0.8964
CariesDetectNet	Group 1	0.7915	0.3253	0.8265	0.1232	0.9423	0.1788	0.6073	0.1233	0.3253
	Group 2	0.8571	0.5301	0.8816	0.2514	0.9615	0.3410	0.7256	0.2514	0.5301
	Group 3	0.9361	0.5180	0.9674	0.5443	0.9639	0.5308	0.7547	0.5443	0.5180
	Group 4	0.9638	0.6024	0.9909	0.8333	0.9707	0.6993	0.8109	0.8333	0.6024
	Group 5	0.9705	0.7590	0.9864	0.8076	0.9820	0.7826	0.8462	0.8076	0.7590
	Group 6	0.9731	0.7590	0.9891	0.8400	0.9820	0.7974	0.8346	0.8400	0.7590
	Group 7	0.9697	0.6867	0.9909	0.8507	0.9768	0.7600	0.8348	0.8507	0.6867
	Group 8	0.9689	0.5783	0.9981	0.9600	0.9692	0.7218	0.7897	0.9600	0.5783
	Group 9	0.9680	0.5783	0.9972	0.9411	0.9692	0.7164	0.7818	0.9411	0.5783
	Group 10	0.9747	0.6626	0.9981	0.9649	0.9752	0.7857	0.8015	0.9649	0.6626

**Table 4 diagnostics-14-02080-t004:** The accuracy, sensitivity, specificity, PPV, NPV, F1-score, Precision, and Recall values of different models in the FTG.

		Acc	Se	Sp	PPV	NPV	F1-Score	Precision	Recall
U-Net	Group 8	0.9800	0.7826	0.9988	0.9854	0.9796	0.87	0.9854	0.7826
	Group 9	0.9820	0.8057	0.9988	0.9858	0.9817	0.88	0.9858	0.8057
	Group 10	0.9921	0.9130	0.9997	0.9968	0.9917	0.95	0.9968	0.9130
	Group 11	0.9838	0.8376	0.9977	0.9730	0.9847	0.90	0.9730	0.8376
	Group 12	0.9835	0.8347	0.9977	0.9729	0.9844	0.89	0.9729	0.8347
YOLOv5n	Group 4	0.9802	0.8289	0.9949	0.9407	0.9835	0.8813	0.9407	0.8289
	Group 5	0.9861	0.8927	0.9952	0.9476	0.9896	0.9194	0.9476	0.8927
	Group 6	0.9784	0.8608	0.9898	0.8918	0.9865	0.8761	0.8918	0.8608
CariesDetectNet	Group 4	0.9680	0.6603	0.9871	0.7608	0.9791	0.7070	0.7608	0.6603
	Group 5	0.9724	0.8490	0.9801	0.7258	0.9905	0.7826	0.7258	0.8490
	Group 6	0.9768	0.8113	0.9871	0.7962	0.9883	0.8037	0.7962	0.8113
	Group 7	0.9757	0.7547	0.9894	0.8163	0.9848	0.7843	0.8163	0.7547
	Group 8	0.9746	0.6603	0.9929	0.8571	0.9793	0.7778	0.8571	0.6603

**Table 5 diagnostics-14-02080-t005:** The accuracy, sensitivity, specificity, PPV, and NPV values of the three models and dental radiologist.

	The Optimal Carious Teeth Number of Train Set	Acc	Se	Sp	PPV	NPV	Youden Index
U-Net	250	0.9921	0.9130	0.9997	0.9968	0.9917	0.9127
YOLOv5n	250	0.9861	0.8927	0.9952	0.9476	0.9896	0.8879
CariesDetectNet	250 or 300	0.9768	0.8490	0.9871	0.7962	0.9905	0.8361
Dental radiologist	-	0.9687	0.9230	0.9725	0.6481	0.9954	0.8955

## Data Availability

The datasets generated during and analysed during the current study are not publicly available due to privacy issues and regulation policies in hospitals but are available from the corresponding author on reasonable request.

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
