# Peer review of "Optimal Training Positive Sample Size Determination for Deep Learning with a Validation on CBCT Image Caries Recognition"

_diagnostics, 2024, doi:10.3390/diagnostics14182080_

Round 1
Reviewer 1 Report
Comments and Suggestions for Authors
The premise of the manuscript is quite interesting and is of immense interest to the dentists.However,the article is very poorly written and the quality of written english needs a major overhaul.A few other points noted are as follows:
1.The authors have not mentioned whether they have obtained an informed consent from the patients for use of data in their study.
2.Have the authors disclosed any conflict of interest in the comparison of the three different software applications?
3.In the methodology section,the authors need to clearly state the standardization measures employed for the acquisition and interpretation of the CBCT images .
4.The authors need to add a note on the types of caries which were interpreted and also whether the carious teeth were anteriors or posteriors and the jaws involved.
5.There are quite a few sentences throughout the manuscript that are quite incoherent and needs to be rephrased for a better understanding.
Comments on the Quality of English Language
The premise of the manuscript is quite interesting and is of immense interest to the dentists.However,the article is very poorly written and the quality of written english needs a major overhaul.A few other points noted are as follows:
1.The authors have not mentioned whether they have obtained an informed consent from the patients for use of data in their study.
2.Have the authors disclosed any conflict of interest in the comparison of the three different software applications?
3.In the methodology section,the authors need to clearly state the standardization measures employed for the acquisition and interpretation of the CBCT images .
4.The authors need to add a note on the types of caries which were interpreted and also whether the carious teeth were anteriors or posteriors and the jaws involved.
5.There are quite a few sentences throughout the manuscript that are quite incoherent and needs to be rephrased for a better understanding.
Author Response
Comment No. 1: The premise of the manuscript is quite interesting and is of immense interest to the dentists. However, the article is very poorly written and the quality of written English needs a major overhaul.
Response:Thank you for the comment. In fact, the manuscript has been revised by a professional proofreading and editing services. Please see the attached Certificate.
Comment No. 2: The authors have not mentioned whether they have obtained an informed consent from the patients for use of data in their study.
Response:Thank you for your valuable comments and thorough review of our manuscript. We acknowledge the importance of ensuring the privacy and confidentiality of patient data. Given that the data used in this study were collected retrospectively, it was not possible to locate all patients to obtain their informed consent an application for exemption from informed consent was submitted to the Ethics Committee and was approved. The application contained a detailed description of the purpose of the study, the methodology, the data protection measures, and why the waiver of informed consent would not adversely affect the rights and health of the subjects. Specifically, we promised that during the study period, the privacy and personally identifiable information of the patients enrolled in the study were kept strictly confidential, and their personal data, such as name, was replaced by code or number, to ensure that no personally identifiable information be disclosed. In addition, should the results of the study publish in a journal, we will strictly comply with ethical norms and ensure that no private information of any patient will be disclosed.
Comment No. 3: Have the authors disclosed any conflict of interest in the comparison of the three different software applications?
Response:Thank you for the question. We have declared in the "Conflicts of Interest" section that "We have no known competing financial interests in the comparison of the three different software applications." This statement ensures that our research is unbiased and free from any potential conflicts of interest that could compromise the integrity of our findings.
Comment No. 4: In the methodology section, the authors need to clearly state the standardization measures employed for the acquisition and interpretation of the CBCT images.
Response:Thank you for the comment. When acquiring the images, the patients were asked to keep the intercuspal position to prevent motion artifact. The patients’ Frankfurt horizontal plane were parallel to the ground. All of the scans performed using the same CBCT unit, and the exposure parameters were strictly chosen according to patient size.
Translucent areas in the teeth were determined to be carious if the following imaging features were met: (1) disruption of the continuity of the enamel margins, large areas of low-density shadows, and areas of reduced density formed by spreading along the enamel dentin boundaries; (2) reduced density of the enamel margins, and low-density shadows confined to the enamel. In addition to the inclusion and exclusion criteria, CBCT images were strictly collected adhering to the diagnostic criteria for caries, and was reviewed by an experienced dental radiologist. This part has been highlighted in section 2.2.
Comment No. 5: The authors need to add a note on the types of caries which were interpreted and also whether the carious teeth were anterior or posteriors and the jaws involved.
Response:Thank you for the valuable comment. Our current study focuses primarily on determining the optimal positive sample size for deep learning in recognizing caries in CBCT images. As such, the input features were designed to capture the morphological characteristics of caries without explicitly classifying them by type, tooth position, or jaw involvement. This aspect, indeed, represents a limitation in our current analysis. In subsequent studies, we will improve the ability of the deep learning network by adding this component so that the deep learning network can not only recognize caries, but also classify a specific type of caries, a specific tooth position (anterior or posterior), and the jaw involved. This enhanced output will undoubtedly improve the clinical utility of our model and its application in dental practice.
Comment No. 6: There are quite a few sentences throughout the manuscript that are quite incoherent and needs to be rephrased for a better understanding.
Response:Thank you for the advice. We have carefully reviewed and revised the draft, especially those expressions that may cause confusion or misunderstanding to readers. In addition, we also sought the assistance of proofreading and editing services provided by Elsevier to ensure that the language quality of the paper meets the high standard required by your esteemed journal.

Reviewer 2 Report
Comments and Suggestions for Authors
After thoroughly examining the presented research entitled "Optimal training positive sample size determination for deep learning with a validation on CBCT image caries recognition" please find below my suggestions to improve the manuscript:
In the Methods section - it would be interesting to explain wheater the dataset used is representative of the wider population or the typical cases encountered in clinical settings. There might be any limitations in data diversity to this matter? How might they affect the results?
In the Results section - are there any additional error metrics that could provide deeper insight into model performance under different conditions. In addition, it would be interesting to provide detailed metrics on the performance of dental radiologists for a more transparent comparison. This includes their sensitivity, specificity, and any other relevant metrics used to assess model performance.
In the Discussion section - please include a rationale for choosing U-Net, YOLOv5n, and CariesDetectNet specifically. Discuss the properties of these models that make them suitable for the task.
Author Response
Response to the comments by the Reviewer 2
Comment No. 1: In the Methods section - it would be interesting to explain wheater the dataset used is representative of the wider population or the typical cases encountered in clinical settings. There might be any limitations in data diversity to this matter? How might they affect the results?
Response:Thank you for your insightful comment and suggestion. The CBCT image datasets used in this study were all from real clinical cases, covering different genders, age groups and different tooth positions with different degrees of caries. The specific data situations have been added and highlighted in the Section 2.2. Although we have tried to cover as many cases of different types as possible during the data collection process, due to the complexity and diversity of the actual clinical situations, some cases located in specific area are still not enough, such as the caries in the anterior teeth. This may lead to a problem in recognizing caries with the deep learning network. This potential diversity limitation may have some impact on the results of the study. We have added this to the limitations at the end of the discussion section.
Comment No. 2: In the Results section - are there any additional error metrics that could provide deeper insight into model performance under different conditions. In addition, it would be interesting to provide detailed metrics on the performance of dental radiologists for a more transparent comparison. This includes their sensitivity, specificity, and any other relevant metrics used to assess model performance.
Response:Thank you for the comment. In order to more comprehensively evaluate the performance of the models under different conditions, in Tables 2, 3, and 4, we added additional error metrics of Precision and Recall, and highlighted in the Results section. In addition to the sensitivity and specificity, we also provide positive predictive value, negative predictive value, accuracy, and kappa value for the dental radiologists’ assessment as well as the Youden index.
Comment No. 3: In the Discussion section -please include a rationale for choosing U-Net, YOLOv5n, and CariesDetectNet specifically. Discuss the properties of these models that make them suitable for the task.
Response:Your suggestion really means a lot to us. The choice of U-Net, YOLOv5n, and CariesDetectNet was based on their proven performance, suitability for the task of CBCT image caries recognition, and their complementary properties that enhance the overall accuracy, precision, and efficiency of our proposed method. The specific content has been highlighted in the Discussion section.

Round 2
Reviewer 1 Report
Comments and Suggestions for Authors
The modifications to the manuscript have been made as per the observations/suggestions.